# Cold Ablation Robot-Guided Laser Osteotome (CARLO^®^): From Bench to Bedside

**DOI:** 10.3390/jcm10030450

**Published:** 2021-01-24

**Authors:** Matthias Ureel, Marcello Augello, Daniel Holzinger, Tobias Wilken, Britt-Isabelle Berg, Hans-Florian Zeilhofer, Gabriele Millesi, Philipp Juergens, Andreas A. Mueller

**Affiliations:** 1Department of Oral and Cranio-Maxillofacial Surgery, University Hospital Basel, University of Basel, 4031 Basel, Switzerland; isabelle.berg@usb.ch (B.-I.B.); hans-florian.zeilhofer@usb.ch (H.-F.Z.); andreas.mueller@usb.ch (A.A.M.); 2Cranio Facial Centre, Hirslanden Medical Center, 5000 Aarau, Switzerland; marcello.augello@hirslanden.ch; 3Department of Oral and Maxillofacial Surgery, Medical University of Vienna, 1090 Vienna, Austria; daniel.holzinger@meduniwien.ac.at (D.H.); gabriele.millesi@meduniwien.ac.at (G.M.); 4Advanced Osteotomy Tools AG (AOT), 4051 Basel, Switzerland; tobias.wilken@aot.swiss; 5MKG Arabellapark, Private Clinic for Oral & Maxillofacial Surgery, 81925 München, Germany; drpjuergens@me.com

**Keywords:** CARLO^®^, robot-guided laser, osteotomy, midface, orthognathic surgery

## Abstract

Background: In order to overcome the geometrical and physical limitations of conventional rotating and piezosurgery instruments used to perform bone osteotomies, as well as the difficulties in translating digital planning to the operating room, a stand-alone robot-guided laser system has been developed by Advanced Osteotomy Tools, a Swiss start-up company. We present our experiences of the first-in-man use of the Cold Ablation Robot-guided Laser Osteotome (CARLO^®^). Methods: The CARLO^®^ device employs a stand-alone 2.94-µm erbium-doped yttrium aluminum garnet (Er:YAG) laser mounted on a robotic arm. A 19-year-old patient provided informed consent to undergo bimaxillary orthognathic surgery. A linear Le Fort I midface osteotomy was digitally planned and transferred to the CARLO^®^ device. The linear part of the Le Fort I osteotomy was performed autonomously by the CARLO^®^ device under direct visual control. All pre-, intra-, and postoperative technical difficulties and safety issues were documented. Accuracy was analyzed by superimposing pre- and postoperative computed tomography images. Results: The CARLO^®^ device performed the linear osteotomy without any technical or safety issues. There was a maximum difference of 0.8 mm between the planned and performed osteotomies, with a root-mean-square error of 1.0 mm. The patient showed normal postoperative healing with no complications. Conclusion: The newly developed stand-alone CARLO^®^ device could be a useful alternative to conventional burs, drills, and piezosurgery instruments for performing osteotomies. However, the technical workflow concerning the positioning and fixation of the target marker and the implementation of active depth control still need to be improved. Further research to assess safety and accuracy is also necessary, especially at osteotomy sites where direct visual control is not possible. Finally, cost-effectiveness analysis comparing the use of the CARLO^®^ device with gold-standard surgery protocols will help to define the role of the CARLO^®^ device in the surgical landscape.

## 1. Introduction

Technological developments in the fields of three-dimensional (3D) printing and virtual surgical planning are producing a shift from conventional surgery to computer-assisted patient-specific interventions. In the field of craniomaxillofacial surgery, virtual surgical planning and the 3D printing of 3D models, cutting guides, patient specific implants, and patient-specific osteosynthesis plates have become part of the daily workflow [1,2,3,4]. In some centers these digital workflows have become the gold standard, thereby stimulating and driving the trend of patient-specific solutions. However, accurately translating these digitally developed preoperative plans in the operating room remains difficult due to the large number of factors involved, including the quality of the imaging data; the software used for segmentation, designing, and planning; the experience of the engineer or surgeon performing the digital planning; the accuracy of the 3D printing or milling process; and, ultimately, the correct intraoperative use of the developed models, guides, implants, and osteosynthesis plates. While the planning is done virtually, the osteotomy is still performed manually and is limited by the physical properties of the osteotome. Multiple studies have demonstrated the clinical relevance and accuracy of a digital workflow, but like with conventional planning, errors and inaccuracies can occur in each step [5]. Furthermore, in virtual planning there are no restrictions on bone movements or theoretical limitations to designs, whereas it is necessary to understand the biological and physical limitations of the surgical instruments and the anatomy in the operating area. The accurate transfer of the preoperatively planned osteotomies to the operating room is reliant on surgical guides that have to be adjusted to the physical properties of the bone cutting instruments, and thus suffer from the same limitations as the surgical instrument used for the osteotomy.

The standard surgical equipment for cutting hard tissue comprises mechanical instruments such as saws, drills, and burs. Due to the inherent bone contact, friction, and pressure, these instruments cause thermal damage to the bone and surrounding tissue, bone fragmentation, and stress in the surrounding tissues [6]. These factors can lead to slower or disturbed bone regeneration or even bone necrosis [7,8]. Moreover, conventional tools are limited by their designs to only being suitable for performing linear or slightly curved osteotomies. Some of these issues have been addressed by the introduction of piezoelectric bone surgery, which induces less vibration in the surrounding tissue and exerts less pressure on the bone surface [6]. Piezoelectric devices can induce micromovements at ultrasonic frequencies, which will simply shake soft tissues while cutting the stiff hard tissues. The advantages of the piezoelectric technique compared with rotary instruments and oscillating saws include higher precision due to better control of the osteotomy line [9,10], faster bone healing and bone regeneration due to less damage to the osteocytes [9,11,12,13], and the preservation of the adjoining soft tissue [9,12].

Lasers have recently been used for cutting hard tissue [14,15]. They can be used to perform contact-free osteotomies and have far fewer geometrical limitations, such as allowing puzzle-like osteotomy designs. However, the accuracy of an osteotomy has remained dependent on the surgical experience and manual precision. Performing a bone osteotomy using a laser offers the potential advantages of high precision, low roughness of cut surfaces, a narrow kerf, and reduced collateral damage to surrounding tissues [7,16,17]. 

An erbium-doped yttrium aluminum garnet (Er:YAG) laser emits radiation at 2.94 μm, and so it can be used for thermal bone ablation due to water exhibiting a strong absorption coefficient at this wavelength. The water molecules selectively absorb the energy, thereby increasing the internal pressure in the form of steam, which causes the explosive destruction of inorganic substances [18]. Potential thermal damage to the surrounding tissues can be practically eliminated by spraying water into the surgical field [6,19]. In contrast to conventional rotating instruments and piezoelectric surgery, laser-induced thermal bone ablation does not create a smear layer on the osteotomy edges. This results in a channeled scaffold that preserves the trabecular ridges, which allows the passage of cells to the site of injury, therefore potentially benefiting bone healing [20,21]. Several preclinical and clinical studies have shown that the healing outcome when using an Er:YAG laser with water cooling is comparable with those for a conventional mechanical osteotomy and piezoelectric surgery [7,22,23,24,25]. A comparison of the healing of a sheep tibia shaft after performing osteotomy between using an Er:YAG laser and a piezoelectric osteotome revealed no relevant macroscopical, radiological, or histological differences after 2 and 3 months [26]. A study comparing bone healing after osteotomies performed with a carbide bur, diamond bur, Er:YAG laser, and pulsed femtosecond laser in mice calvaria also found no significant differences in bone analyses performed with micro computed tomography (micro-CT) after 12 weeks [27].

In order to overcome the geometrical limitations of conventional bone cutting instruments, avoid the reliance on the manual skill of the surgeon, and mitigate the difficulties of translating the digital preoperative planning to the operating room, a Swiss start-up company (Advanced Osteotomy Tools [AOT]) has developed a miniaturized robotic laser system. A laser head was mounted on a robotic arm guided by intraoperative navigation to produce the Cold Ablation Robot-guided Laser Osteotome (CARLO^®^). The goal was to develop a stand-alone robot-guided laser for performing contact- and debris-free osteotomies, eliminating the need for cutting guides and manual maneuvers by surgeons. This device offers the possibility of a total digital workflow that allows the direct transfer of the virtual planning into the operating room. The potential of the CARLO^®^ device has been confirmed previously in in vivo animal studies that compared piezoelectric surgery and the CARLO^®^ device in minipig mandibles and sheep skulls, which the results showing that the osteotomies had similar durations but higher accuracies and a tendency of faster bone healing when the CARLO^®^ device was employed [25,28,29]. The applicability of the CARLO^®^ device for intraoperative use for craniomaxillofacial interventions has been demonstrated on human cadavers by successfully performing mandible split osteotomies, fibula osteotomies, and midface osteotomies with various designs [14].

In 2019, the first-in-man clinical study to evaluate the safety and accuracy of linear midface osteotomies performed with the CARLO^®^ device was started (ClinialTrials.gov Identifier: NCT03901209). The goal of this study is to assess the safety and accuracy of the CARLO^®^ device, the results of which will be published in another article. Herein we explain the workflow and our practical experience when using the CARLO^®^ system, as exemplified by its application to one of the study patients.

## 2. Methods and Results

### 2.1. Ethics

The study was approved by the Swiss Agency for Therapeutic Products (Swissmedic Ref 10000471) and the ethical committee for Northwest and Central Switzerland (Ethikkommission Nordwest- und Zentralschweiz [EKNZ] 2017-01768). The study was registered with the European Database on Medical Devices (EUDAMED CIV-19-02-027204). The patient signed a written informed consent sheet for study participation.

### 2.2. CARLO^®^ Device

The CARLO^®^ device includes an Er:YAG laser mounted on a robotic arm that is guided by an integrated navigation system (Figure 1). The laser produces a beam with a focal diameter of 0.8 mm, a pulse energy of >100 mJ, and a pulse rate of 10 Hz. The laser is operated in the cold ablation regime, which leaves the ablated bone surface porous and biologically functional since the temperature of the surrounding tissue never exceeds 45 °C. This can be achieved by ensuring that the duration of each laser pulse is sufficiently short, sweeping the laser head along the osteotomy line, and providing constant cooling and humidification from the saline solution that is sprayed from a nozzle. A visible green laser is coaxially aligned with the cutting beam of the Er:YAG laser to allow the surgeon to monitor and simulate the osteotomy path prior to the cutting. The robotic arm (lightweight medical grade robot LBR Med, KUKA, Augsburg, Germany) on which the laser head is mounted provides a lateral repeatability of better than 0.15 mm and an angular repeatability of better than 7 mrad. The robotic arm was designed to be tactile and to move at safe speeds and exert forces compatible with human–machine interactions. The arm will move to a predefined safe position when it detects the possibility of a collision. The robot is guided by a navigation system based on two tracking cameras that observe two markers: one integrated in the laser head and the other attached to the patient. A nozzle that sprays normal saline solution is used to cool, moisturize, and capture most of the ablation debris around the targeted tissue. 

### 2.3. Digital Workflow

A 19-year-old male patient in good general health but presenting a hypoplastic maxilla and a history of a soft palatal cleft provided written informed consent to undergo a linear Le Fort I midface osteotomy performed with the CARLO^®^ device. Virtual planning was based on preoperative CT (SOMATOM Definition Flash, Siemens Healthcare GmbH, Erlangen, Germany) with a slice thickness of 1.0 mm performed 5 weeks prior to the surgery. The midface was segmented using Mimics software (version 21.0, Materialise, Leuven, Belgium). The dental surfaces and bite registration were obtained with an intraoral scanner (TRIOS 3 Basic, 3Shape, Copenhagen, Denmark) and aligned with the dental surface of the segmented midface. Two linear Le Fort I osteotomy lines were designed in Mimics that took into account the positions of the dental roots (Figure 2). The virtually planned Le Fort I osteotomy was exported as a Surface Tessellation Language (STL) file and then imported into the software of the CARLO^®^ device, which includes a graphical user interface. Since the osteotomy line can be adjusted within the software, intraoperative corrections can be performed without the need to return to the planning software. The cutting angulation of the laser head relative to the bone surface was set using the graphical user interface, and this can also be adjusted intraoperatively. The orthognathic virtual surgery planning was finalized in IPS CaseDesigner (KLS Martin, Tuttlingen, Germany), which allows the in-house manufacture of an occlusal splint using a 3D printer (Objet30 Prime, Stratasys Ltd., Eden Prairie, MN, USA).

### 2.4. Surgical Procedure

The intraoperative procedure included the usual sterile draping of the patient and obtaining surgical access with mucoperiosteal flap elevation in the maxilla. The registration marker was registered in the software of the CARLO^®^ device using the optical cameras and navigation software, and subsequently attached to the hard palate using two self-drilling 11.0-mm intermaxillary fixation (IMF) screws (M5248.11, Medartis AG, Basel, Switzerland). The marker was positioned in direct sight of the optical cameras on the left side of the patient so as to not interfere with the surgeon, who was positioned on the right side of the patient (Figure 3). Point-pair matching of at least four landmarks was used to match the virtual maxilla with the maxilla of the patient (Figure 4). Dental landmarks and the infraorbital foramina were used for the registration process, which took 5 min and resulted in a root-mean-square (RMS) error of 1.0 mm.

Two black malleable orbital retractors were placed behind the tuber maxillae and apertura piriformis in order to retract the soft tissues and avoid scattering of the laser beam (Figure 5). The operating team, nursing staff, and patient were all equipped with protective laser goggles. The laser head is automatically positioned in the operating area by the robotic arm and the osteotomy line is visualized by the green noncutting laser beam mentioned above. This step is approved by the surgical team. In the exemplar case, the surgical team was not satisfied with the on-site osteotomy line visualization, and so the osteotomy line as well as the laser head angulation were adjusted intraoperatively (Figure 2). Once the procedure is agreed, the laser needs to be manually turned on and controlled by the surgeon by maintaining continuous pressure on a trigger button. The robot-guided laser then performs the osteotomy autonomously while still under direct visual control by the surgical team, who are provided with real-time visualization of the cutting area on a monitor and real-time auditory feedback. The amount of bone removed during the osteotomy is displayed in a histogram on the graphical user interface, which allows for depth control. Once the osteotomy of the inner cortex of the bone has been completed, the sound of the laser cut changes, informing the surgeon that the osteotomy is complete, as demonstrated in the attached video (Appendix A).

The osteotomy took approximately 8 min on the right side and 4 min on the left side. Multiple safety protocols were in place to address any unexpected events, such as laser-stop buttons on the device held by the surgeon and on the robot itself, and the ability to push the robotic arm away. The CARLO^®^ device successfully performed the linear midface osteotomy in the present case on both sides without encountering any technical problems or safety concerns. After the osteotomy had been performed, the CARLO^®^ device was removed and the osteotomy of the nasal septum, lateral nasal wall, and pte-rygomaxillary junction was completed using a chisel and hammer. The surgery was then completed in a conventional manner.

### 2.5. Postoperative Procedure

Postoperative CT with a slice thickness of 1.0 mm was used to compare the performed and planned osteotomies by superimposing the preoperatively planned and osteotomized maxillae in Mimics (version 21.0). The virtual model with the intraoperatively adjusted osteotomy line on the right maxilla was exported as an STL file from the CARLO^®^ software and imported into Mimics. A maxillofacial surgeon measured the distances from four predefined dental surfaces to the osteotomy line in both models. In order to ensure that the measurement positions were correct, the pre- and postoperative models were superimposed based on the dental surface scan. Laterally, the postoperatively mea-sured distances from the mesiobuccal cusps of the right and left first upper molars to the lateral osteotomy line showed differences of 0.03 mm and 0.79 mm, respectively, relative to the preoperative planned distances. Anteriorly, the postoperatively measured distances from the right and left first incisors to the medial osteotomy line showed differences of 0.51 mm and 0.45 mm, respectively (Figure 6). The patient was discharged on the third postoperative day and showed normal healing at 1-, 2-, and 4-week follow-ups, with no reports of infections, mucosa dehiscence, or hematoma. There was no reported adverse event related to the use of the CARLO^®^ device.

## 3. Discussion

The CARLO^®^ device essentially represents the fusion of three preexisting technologies: lasers, robotics, and navigation. It includes a miniaturized Er:YAG laser head placed on a robotic arm that is guided by a point-based optical navigation system. Manually operated Er:YAG lasers have been used successfully in the past for performing intraoral bone osteotomies with different indications [7,30,31], and point-based optical navigation technology is well established for surgical planning [32]. The precision and accuracy of the CARLO^®^ device has previously been tested preclinically in small cadaver and animal studies, which showed clinically acceptable accuracy and bone healing compared with piezoelectric surgery [14,25,28,29]. The CARLO^®^ device has subsequently been tested for its usability in performing Le Fort I osteotomies in human cadavers. These data are unpublished, but they have been disclosed to the Swiss Agency for Therapeutic Products. Based on these cumulative results, the CARLO^®^ device was deemed applicable and safe for the first-in-man clinical testing performed in the present study under direct visual control.

For the first-in-man use of the CARLO^®^ device, we opted for a linear Le Fort I osteotomy since this is a highly standardized and preplanned procedure. Osteotomies of the nasal septum, lateral nasal wall, and pterygomaxillary junction were excluded from the study protocol for safety reasons, since the accuracy of the device has not yet been confirmed in a clinical setting and direct visual feedback would not have been available du-ring these procedures. The eventual goal is to perform all osteotomies using the CARLO^®^ device without the need for conventional instruments, but this has not been achieved yet. From a technical perspective, it should be possible to perform any kind of osteotomy in the human body as long as the laser beam can reach the site of osteotomy when any intervening soft tissues have been sufficiently retracted. The present anterior maxillary osteo-tomy did not need any additional assistance from conventional mechanical instruments, whereas the osteotomies of the lateral nasal wall, nasal septum, and pterygomaxillary junction were performed using a chisel and hammer in accordance with the study protocol. The intraoperative safety of the patient, operating team, and nursing staff was gua-ranteed by equipping them with laser goggles and providing the surgeon with direct access to multiple emergency laser-stop buttons and a graphical user interface with real-time visual and auditory feedback from the surgical site. However, if the CARLO^®^ device is intended to be used in the future at sites where direct visual feedback on its performance is impossible, such usage will need to be reassessed in a preclinical setting before being applied to patients.

The present patient showed uneventful healing during an 11-month follow-up period, with radiologically confirmed complete bone healing of the osteotomy line. There were no intra- or postoperative complications related to the CARLO^®^ device. The absence of a smear layer, intact trabecular structure, and absence of thermal bone alteration along the osteotomy line should theoretically lead to improved healing conditions due to the lack of disturbance to cell migration into the osteotomy gap [21]. Most of the studies in the literature used animal subjects and histological or radiological parameters to analyze bone healing [6]. Based on the current literature, we can assume that performing a bone osteotomy with a water-cooled Er:YAG laser results in similar histological and radiological bone healing compared with using conventional rotating instruments and piezosurgery [6,25,26,27,29]. A more-pronounced bone bleeding with a subsequent risk of hematoma formation could be presumed when using the Er:YAG laser due to the absence of a mechanical smear layer along the osteotomy bone surface. In the present patient, both the intraoperative bone bleeding and postoperative swelling were within the normal clinical ranges, but further clinical research is mandatory to compare the degrees of bone bleeding and bone healing between mechanical and laser osteotomies.

The accuracy of the CARLO^®^ device is affected by the performance of the navigation system. In the present case, the point-based registration achieved high accuracy. However, it is prone to error since the resolution of the CT data set influences the registration process and the identification of anatomical landmarks on the segmented image and patient [32]. When using dental landmarks as registration points, it is crucial to align the initial segmentation of the midface with dental surfaces derived from an intraoral surface scan, since dental surfaces derived from CT typically have lower spatial accuracy. In addition to the dental landmarks, the infraorbital foramina provided reliable landmarks during the registration process, resulting in high overall registration accuracy. It is also important to take sufficient time when fixing the target marker and registering the patient’s skull with the pointer tool in order to achieve an RMS error of ≤1.0 mm, which has been described as acceptable in the literature [32]. This is an essential step in the successful utilization of the CARLO^®^ device. The importance of high-quality thin-slice imaging must also be highlighted in the preoperative planning and postoperative analysis. Cone-beam computed tomography (CBCT) is prone to interference artifacts from orthodontic metal appliances, requiring their manual removal during the segmentation process. This labor-intensive and time-consuming procedure can result in inaccuracies during the subsequent surgical planning. Therefore, although CT results in higher radiation exposure for the patient, it is preferred over CBCT to provide optimal preoperative planning and accurate registration of the navigation system.

Postoperatively, the maxilla with the adjusted osteotomy line was directly exported from the CARLO^®^ software as an STL file, allowing for pre- and postoperative comparisons of the osteotomy accuracy. Superimposing the virtually planned maxilla with the adjusted osteotomy line and the postoperative maxilla segmented from postoperative CT revealed a clinically acceptable maximum difference of 0.8 mm for the left posterior measurement point, with an RMS error of 1.0 mm. It should be possible to achieve higher accuracy by reducing the RMS error, since Augello et al. reported achieving maximum RMS errors of 0.14 mm [14].

Registering the target marker took 5 min in the present study. The linear osteotomy performed using the CARLO^®^ device took approximately 8 min on the right side and 4 min on the left side in the present case. While this is clinically acceptable, it is also necessary to take into account the times needed for fixation of the target marker, for the robot to position itself, and for the visualization of the osteotomy line. These parameters were not included in the study protocol, but the times taken can be expected to reduce with experience. 

During the present first-in-man use of the CARLO^®^ device, we identified a number of aspects that should be further improved. For the present patient the target marker was attached to the hard palate using two 11.0-mm self-drilling IMF screws. Placing a marker could be difficult in patients with reduced mouth opening and lead to the lower lip or mandible exerting undesired pressure on the marker during surgery. The use of an optical navigation system means that the marker needs to be directly viewable by the tracking cameras. Combined with the proximity of the laser head to the patient’s skull, this resulted in a crowded operating area that complicated the retraction of soft tissue during the osteotomy. Therefore, the location of the marker and its means of fixation need to be optimized. Positioning the marker on the temporal skull using a head band could ensure direct line-of-sight between the cameras and the marker, allow for an assistant to stand on the side of the marker, and avoid interference with the laser head as well as the need to place extra (palatal) screws. Also, when attempting bimaxillary surgery with the CARLO^®^ device, it is likely that the target marker will need to be repositioned on the mandible to perform the sagittal split osteotomy, resulting in a second registration process. An innovative solution therefore needs to be developed to reduce the operating time.

The CARLO^®^ device provides auditory and visual feedback about the laser that helps the surgeon to control the osteotomy being performed. However, there is no active depth control, meaning that the laser will keep cutting even after the osteotomy has been completed. In the current version of the device, it is therefore mandatory to protect the underlying tissue with a laser-beam absorber, which in our case was achieved by placing two black malleable orbital retractors behind the tuber maxillae and apertura piriformis. However, these retractors can only partly protect the deeper tissues from the laser beam since the posterior wall of the maxillary sinus cannot be reached. An active feedback system could be implemented to stop the laser cutting once the osteotomy has been completed or once only a very small amount of bone tissue remains, which then can easily be removed manually.

The water-cooling system prevents thermal damage to the bone but creates a spray that reaches the entire operating field, and hence is likely to also affect the operating team due to the aforementioned crowded operating area. This could be optimized by adjusting the nozzle or implementing protecting shields, thereby making the device safer for the operating team. 

The new system should ideally be compared with the established surgical protocols in order to evaluate its clinical relevance and cost-effectiveness. The CARLO^®^ device might have specific benefits for use in so-called waferless orthognathic surgery [33]. In this technique, the final position of the maxilla relies on an exact match between preplanned osteotomy lines and the personalized prefabricated internal fixation plates. This can reduce the preoperative planning steps, since the virtual planning can be exported from any commercially available software and directly imported as an STL file into the software of the CARLO^®^ device. This means that there is no need to design and manufacture cutting guides, potentially reducing time and cost. The elimination of the guides additionally avoids issues that might arise from inaccuracies in the design of the cutting guide, or the interference of soft tissue preventing a perfect fit. On the other hand, when comparing the system with virtually planned orthognathic surgery performed using 3D printed wafers without cutting guides, additional planning time could be expected, especially when using CBCT, since extra time would be needed to remove scattering over the osteotomy line. 

Other potential benefits of the direct transfer of data from virtual planning to the CARLO^®^ device can be found in head and neck reconstruction surgery, where any delay from diagnosis to surgical resection can have detrimental effects on the patient [34].

A cost-effectiveness analysis of the present system was not performed, and so its potential advantages will have to be weighed against expected extra costs arising from the system itself and extra surgical time for the installation of the robot, marker fixation, and registration of the navigation system. 

A technique for minimally invasive orthognathic surgery has been described, in which the maxillary bone is exposed by making a small mucosal incision from tooth 12 to tooth 22, osteotomized with an oscillating saw or piezoelectric osteotome, and downfractured by combining manual manipulation with using a chisel and hammer. This minimally invasive technique appears to reduce postoperative morbidity and the length of hospital stay [35,36,37]. Therefore, the applicability of the CARLO^®^ device as an alternative to conventional osteotomy in this minimally invasive technique should be assessed in a future cadaver study. 

The robot-guided laser osteotome shows clear potential in maxillofacial surgery, where high precision is necessary to achieve optimal functional and esthetic outcomes. In the present case, the system accurately transferred from preoperative planning to the operating room, without the need for a cutting guide. Keeping the RMS error as low as possible (preferably <1.0 mm) is crucial for successful outcomes. Due to the ability to perform a contact-free osteotomy, the range of osteotomy designs is limited only by the imagination of the surgeon and by human anatomy. Combined with its bone healing capacity being comparable to that of piezoelectric surgery [7,22,23,24,25], we believe that the CARLO^®^ device has the potential to become an important new tool in all bone-cutting disciplines (Table 1). This device could be employed in a wide range of bone osteotomies over the entire body if sufficient bone exposure is possible. Its freedom in cutting designs will provide the creative surgeon with the means to treat every patient individually and will probably also help in identifying new approaches to existing surgical techniques.

## 4. Conclusions

The CARLO^®^ device could be a useful alternative to conventional osteotomy instruments and piezoelectric surgery for bone osteotomies. In the present exemplar orthognathic case, the linear part of the Le Fort I midface osteotomy was safely completed with this newly developed autonomous system, with no technical problems being encountered. The CARLO^®^ device offers the advantage of contact-free osteotomies, theoretically unlimited osteotomy geometries, and the capability to directly transfer from preoperative planning to the operating room without the need for cutting guides. However, performing osteotomies where direct visual control is not possible will require the implementation of an active depth control system and further preclinical studies confirming the safety and accuracy. Furthermore, the impact on bone healing needs further assessment, particularly compared with conventional osteotomy instruments. The present results combined with tests of further specific applications for the CARLO^®^ device in craniomaxillofacial and orthopedic surgery will clarify and define the role of this novel device in the surgical landscape.

## Figures and Tables

**Figure 1 jcm-10-00450-f001:**
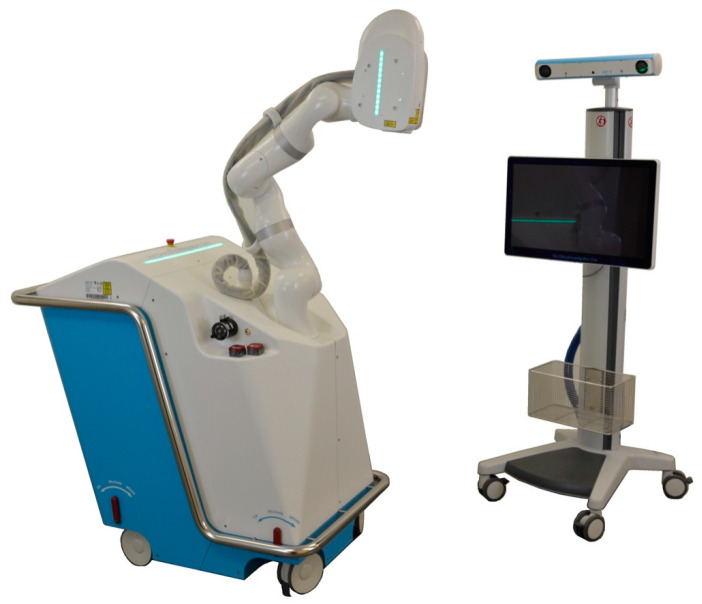
Cold Ablation Robot-guided Laser Osteotome (CARLO^®^) developed by Advanced Osteotomy Tools (AOT AG, Basel, Switzerland) consisting of a laser head mounted on a robotic arm and guided by an optical navigation system.

**Figure 2 jcm-10-00450-f002:**
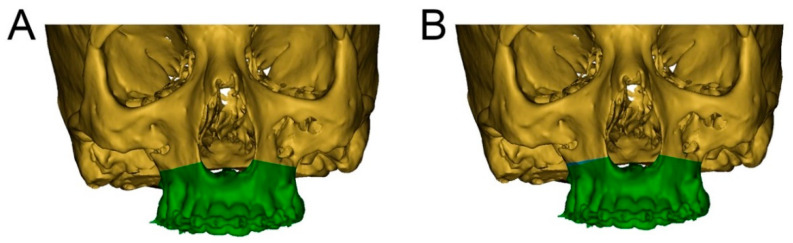
Segmentation of the cranium (green) and maxilla (brown) with preoperatively planned and intraoperatively adjusted osteotomy lines. (**A**): Preoperatively planned left and right linear midface osteotomy lines in Mimics. This model is exported as an STL file and then imported into the software of the CARLO^®^ device. (**B**): Intraoperatively adjusted osteotomy line on the right side of the maxilla, marked in blue.

**Figure 3 jcm-10-00450-f003:**
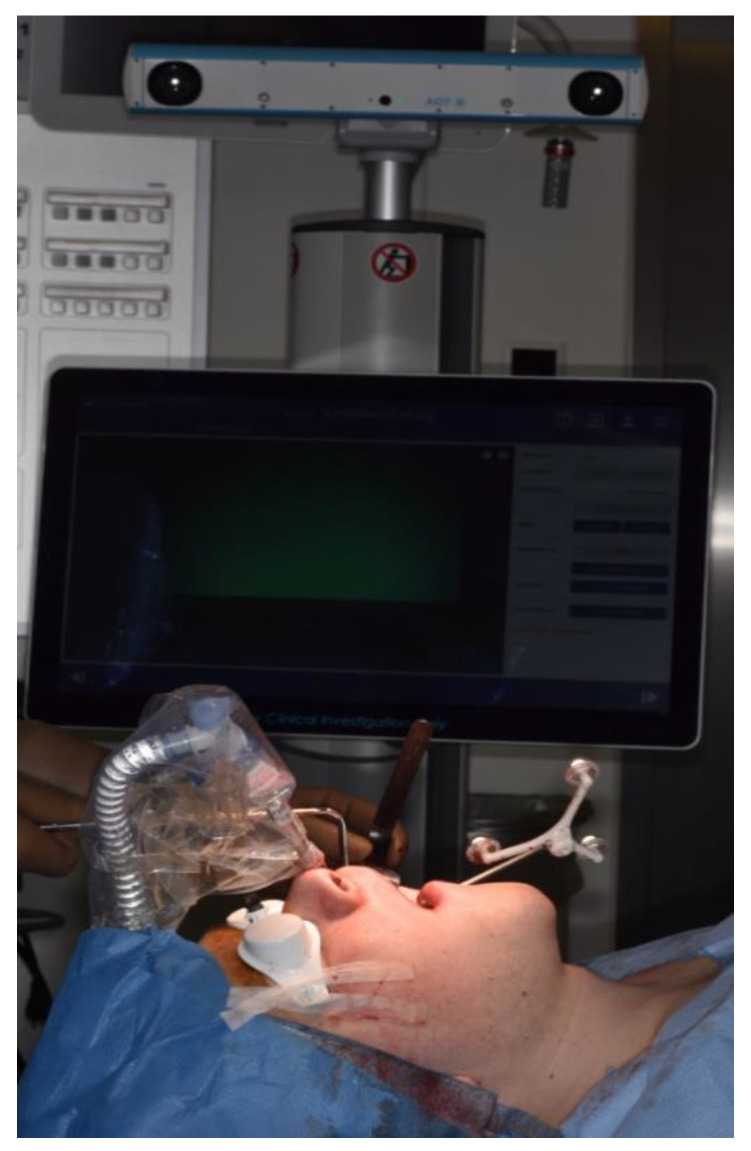
Intraoperative position of the CARLO^®^ navigation system on the left side of the patient with the navigation marker attached to the hard palate using two 11.0-mm self-drilling intermaxillary fixation (IMF) screws. The white goggles provide ocular protection during laser cutting. The graphical user interface on the left side of the patient provides real-time visual information about the operating area obtained by a camera embedded in the laser head and about the bone thickness in the form of a histogram. Auditory feedback informs the surgeon when the osteotomy is completed.

**Figure 4 jcm-10-00450-f004:**
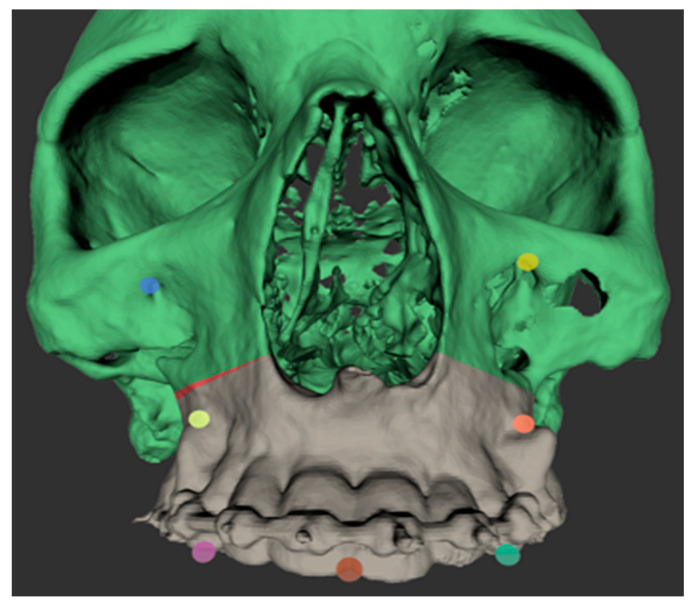
Screenshot of the registration points used for point-based registration of the patient’s skull to the digital model imported into the software of the CARLO^®^ device. The osteotomy line on the right maxilla can be adjusted intraoperatively directly in the software of the CARLO^®^ device; the adjustment made is shown in red.

**Figure 5 jcm-10-00450-f005:**
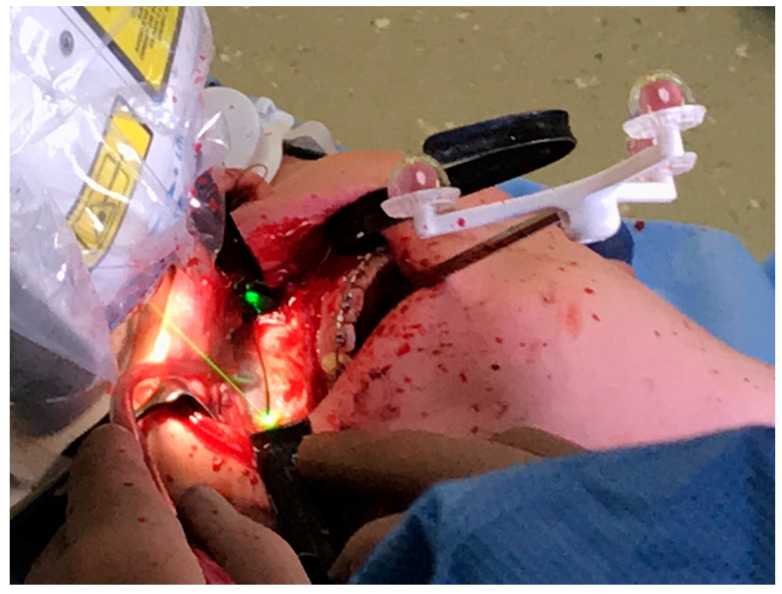
Intraoperative photograph obtained during a linear Le Fort I osteotomy of the right maxilla performed with the CARLO^®^ device. Soft tissue was retracted using two black malleable orbital retractors placed behind the tuber maxillae and apertura piriformis. The registration marker was attached to the palate using two 11.0-mm self-drilling IMF screws and positioned in direct sight of the optical camera system on the left side of the patient. The coaxially oriented noncutting green laser beam indicates the location of the osteotomy line.

**Figure 6 jcm-10-00450-f006:**
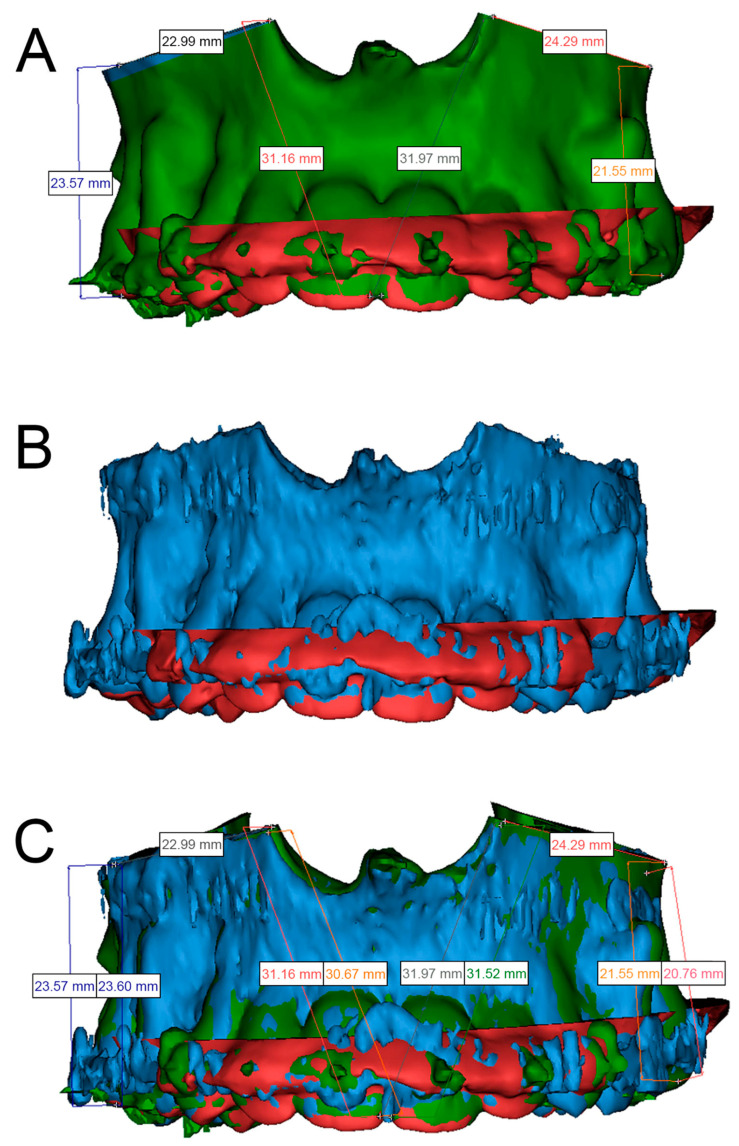
Superimposed preoperative (green) and postoperative (blue) segmented maxillae based on the intraoral dental surface scan (red). (**A**): Preoperative maxilla with an intraoperatively adjusted osteotomy line that was imported into Mimics. The intraoperatively adjusted osteotomy line on the right side of the maxilla is shown in blue, and the aligned dental surface scan is shown in red. The measured distances from the mesiobuccal cusps of the left and right upper molars to the lateral osteotomy line and from both incisors to the medial osteotomy line are shown. (**B**): Postoperative maxilla segmented in Mimics aligned with the dental surface scan shown in red. (**C**): Superimposition of the two maxillae based on the aligned dental surface scan. The measurements for each landmark are presented in pairs, with those on the left and right representing the planned and performed osteotomies, respectively. The difference was 0.03 mm from the mesiobuccal cusp of tooth 16 to the right lateral osteotomy line, 0.51 mm from the mesial surface of tooth 11 to the right medial osteotomy line, 0.45 mm from the mesial surface of tooth 21 to the left medial osteotomy line, and 0.79 mm from the mesiobuccal cusp of tooth 26 to the left lateral osteotomy line.

**Table 1 jcm-10-00450-t001:** Advantages and limitations of the current CARLO^®^ device.

Advantages	Limitations
Use of robotics with an integrated navigation system results in an autonomously performed hard tissue osteotomy that eliminates the need for manual maneuvers by surgeons	Need for greater surgical exposure of the operating area to allow for direct visual control of the osteotomy
2. Contact-free osteotomy achieved using an Er:YAG laser avoids geometrical limitations faced by conventional methods	2. Extra anesthesia time for the installation of the CARLO^®^ system and fixation and registration of the marker
3. Translation of the digital preoperative plan into the operating room eliminates the need for a cutting guide	3. No active depth control system, with depth control primarily based on auditory feedback
4. Intraoperative adjustments of the planned osteotomy line are possible using the software of the CARLO^®^ device	4. The accuracy of the osteotomy is highly dependent on the quality of the registration performed by the surgical team
5. Presumed beneficial bone healing due to the preservation of the trabecular bone structure and the absence of a smear layer and thermal damage	5. Unclear risks of bleeding and hematoma formation due to the absence of a smear layer and thermal damage

## Data Availability

The data presented in this article are available within this article and the Appendix A.

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
