# Peer review of "Cold Ablation Robot-Guided Laser Osteotome (CARLO®): From Bench to Bedside"

_jcm, 2021, doi:10.3390/jcm10030450_

Round 1

Reviewer 1 Report

Excellent presentation of the findings; clinical relevance is tough for me a I am not an oral surgeon but this potentially could be applicable to an orthopaedic population. Intriguing findings.

Author Response

Dear Sir or Madam,

Thank you very much for taking time to read and comment on our submitted article. We are very glad about the possibility to submit a revision of our work and tried our best to meet the reviewers’ criteria.

Thank you for your very positive feedback. If all goes well, the device should confirm the accuracy and safety in a bigger patient cohort. Indeed, the device should be applicable for all osteotomies in the body if enough surgical exposure can be provided. However, research into cost-effectiveness and clinical relevance will be necessary.

We wish you nice holidays.

Kind regards,

Matthias Ureel

Reviewer 2 Report

Brief summary
This paper describes the usage of a (commercial) robot laser guiding system in an orthognathic surgical case. The preoperative and perioperative workflow are presented as well as the authors first experiences of this innovative setup in a clinical case belonging to a larger study sample. A postoperative evaluation of the result is presented.

Broad comments
Overall the technique described in this paper is most innovative and very interesting. Using a robot-guided laser to execute a preplanned osteotomy is inventive and a case report describing its usage in a clinical case is unique. There are, on the other hand, some majors concerns to be noted:
- the overall impression this paper gives on the performance of different aspects of the systems seem to lack a strong critical note. Despite several relevant disadvantages of the system are given in the discussion section it seems to heavily promote the advantages of the system. Some advantages that are stated, for example in table 1, lack a strong scientific base.
- the clinical advantages seem to be overly promoted. A few examples: in what way is the replacement of a simple cutting guide for an intra operative navigation beneficial? The system, for now, requires extra surgical exposure, the placement of extra screws on the palate and the placement of a registration marker very close to the surgical field? In what way does the precise placement of an osteotomy line influences the postoperative results for a patient, which seems to be largely determined by achieving the desired displacement? How does a direct line of sight up until the tuber maxillae weighs against a modern ‘minimal invasive’ approach to the Le Fort I osteotomy? Mal- or nonunion of an osteotomy of the midface is a (very) rare complication, in what way (and on what grounds) do the authors expect a laser osteotomy to improve the healing process from the patients perspective? Especially since the laser is only used for the outer lining of the Le Fort I osteotomy and conventional instruments are used for osteotomies of the lateral nasal wall, nasal septum and pterygoid-maxillary junction. Uncertainties like these should be addressed when discussing first clinical usage.
- some relevant details of the workflow are lacking, for example, the placement of a laser beam absorber behind the osteotomy line to protect the underlying tissue. How is this step performed and is it safe in the more difficult to reach areas such as the pterygoid region as proposed in line 248? The osteotomy line has changed during the procedure. Why was it changed and how was this taken into account when comparing the preoperative plan with the postoperative osteotomy? In figure 5 it seems that the lower lip exerts some pressure on the registration star, are there any relevant concerns towards this setup in specific groups of patients? The Le Fort I osteotomy involves more than a straight line osteotomy of the visible parts of the maxilla. It can be considered standard practice to also clear the lateral nasal wall and bony nasal septum but only the pterygoid-maxillary junction is mentioned in the workflow and discussion.
- the accuracy of this setups strongly depends on the performance of the navigational system as is stated in the discussion section. A study in which the systems accuracy and precision is validated in, for example, a cadaver study seems not to be present.

Specific comments
- the advantages as stated in table 1 seem to be insufficiently supported by this paper or its references
- the references 6,7,8 in line 58-62 provide a one sided view and insufficient support for the statements made in the same paragraph. For example, including the study performed by Martin Cloutier et al (Calvarial bone wound healing: a comparison between carbide and diamond drills, Er:YAG and Femtosecond lasers with or withouth BMP-7) would show a slightly more nuanced comparison between mechanical and laser bone ablation.
- line 200, no unexpected event occurred. This seems not to be in line with any last minute changes to the osteotomy line during surgery. 
- Safety of this setup is indeed important, but future research into the cost effectiveness of adding a laser to this surgery has not been mentioned at all. Whilst this is not of utmost importance in this stage of innovation, it should be addressed, at least. Extra costs might be expected due to the costs of the system itself, the extra surgical time necessary for exposure and registration as well as any extra time spend on preparing the case digitally (including removal of any scattering on the osteotomy line, see line 278)
- 286 a direct line of sight to a pterygoid-maxillary junction (see line 248) seems infeasible in some  cases due to strong anatomical variations. The question arises how this weighs against a minimal approach with piezo surgery.

Author Response

Dear Sir or Madam,

Thank you very much for taking time to read and comment on our submitted article. We are very glad about the possibility to submit a revision of our work. We took the liberty to rearrange and combine some of the reviewers’ questions as some answers are linked to different questions. We tried our best to meet the reviewers’ criteria.

We wish you very nice and healthy holidays.

Yours Faithfully,
Matthias Ureel

Reviewer 3 Report

Congratulations on this study introducing a very impressive and potentially helpful robotic device.

This article provides excellent description of the procedure, including all the relevant details for planning and executing the surgery.

As described in the introduction, the aim of this paper is the description of the workflow and practical experience with the CARLO robotic device in one study patient. Further evaluation regarding safety and accuracy is expected with the publication of the clinical study, as mentioned.

In the abstract, the discussion as well as the conclusion, the author presents the method as safe and accurate. It is not appropriate to draw that conclusion after the presentation of one case only.

Also, for surgeons interested in purchasing the device, an estimation of the time required for planning and the overall procedure time compared to established methods would be helpful.

Author Response

Dear Sir or Madam,

Thank you very much for taking time to read and comment on our submitted article. We are very glad about the possibility to submit a revision of our work. We tried our best to meet the reviewers’ criteria.

Thank you for your positive feedback. Below you can find our point-by-point replies and adjustments in the article. In general, we agreed that all comments were absolutely justified and have aided in improving the quality of our work. For that, we thank you. 

Congratulations on this study introducing a very impressive and potentially helpful robotic device.

This article provides excellent description of the procedure, including all the relevant details for planning and executing the surgery.

As described in the introduction, the aim of this paper is the description of the workflow and practical experience with the CARLO robotic device in one study patient. Further evaluation regarding safety and accuracy is expected with the publication of the clinical study, as mentioned.

In the abstract, the discussion as well as the conclusion, the author presents the method as safe and accurate. It is not appropriate to draw that conclusion after the presentation of one case only.

We agreed, in our enthusiasm we have made an unfounded generalized conclusion. In general, the discussion lacked a more critical analysis of the device, especially concerning safety and accuracy but also cost-effectiveness and clinical relevance. These aspects have been changed throughout the text and in the abstract and conclusion. The discussion has been re-written to a more critical reflection of the CARLO system.  

Also, for surgeons interested in purchasing the device, an estimation of the time required for planning and the overall procedure time compared to established methods would be helpful.

Indeed, we agree. Unfortunately the time necessary for the virtual planning has not been measured. However, compared to standard virtual surgical planning cases there will be a reduction of planning time as there is no need to design cutting guides to translate the digital planning to the operating room. Also, the segmented models can be directly imported as STL files into the CARLO software. We have added a paragraph describing these remarks to the discussion.

As this was the first use of the device, we didn't compare the total procedure time to established procedures. A clinical study with a bigger patient cohort should give an indication of the average procedure time compared to established surgery protocols.

We hope you are satisfied with our proposed adjustments and wish you very nice holidays.

Yours Faithfully,
Matthias Ureel

Round 2

Reviewer 2 Report

Brief summary of comments.
The paper has been significantly improved on its critical attitude and presentation of the experiment. Whilst I strongly agree with my fellow reviewers that the presented case report and described technology is impressive and unique, there are still some (very) relevant concerns in my opinion. I do believe that the results of this first use case deserve publication, but overall the impression the paper gives to the capabilities of the system seems not to be well in line with the actual presented data.
One of the major concerns is the lack of preclinical studying of the proposed system and the unwillingness (of the company) to share data on this part. Secondly, the paper (still) seems to somewhat overstate its capabilities, for example, claims on its precision in the (abstract and) result section should be weakened (see further down for an explanation). Last but not least it is mentioned a few times that the benefits of the system should be further explored in new patient series whilst there are still significant issues that can and should be addressed before testing the system on actual patients in the near future, in example:
- active depth control or laser beam absorber for parts that cannot be protected with retractors
- preclinical studying of its performance in regions without visual feedback
- preclinical studying of its precision in regions with visual feedback
- the issues with the target marker design (especially for the lower jaw)
These shortcomings should be made more clear, because the abstract still reads as if the robot successfully performed a ‘linear Le Fort I osteotomy’ with an ‘accuracy of 0.8mm’ which is ready for ‘further research in a larger patient sample’.

More detailed comments, in reference to the cover letter:
Page 1
- the suggested adjustment to table 1, adding disadvantages, is valuable
- minor comment: ‘there is no need to design and manufacture cutting guides, thus reducing time and costs’. I understand the upside and the possibilities provided by the system, but without any scientific grounds to stand on, I would advice to correct this to express uncertainty in terms of actual timesaving or cost-effectiveness, especially against the ‘gold standard’ which is, to my opinion, not the usage of patient specific cutting guides. The time and costs associated with the steps between confirming a regular virtual planning by the surgeon and getting a wafer (or even simple cutting guide) prepared by the (department) 3D-printer, for example, are not as significant.

Page 2
- the addition of the uncertainties surrounding the target marker is valuable
- minor comment: I would advise to rephrase the part explaining the exclusion of some of the structures to the laser osteotomy: ‘the accuracy of the device has not yet been confirmed in a clinical setting’. As described elsewhere in the rebuttal, the accuracy of the device in preclinical conditions has not been published and without direct visual feedback, an osteotomy here is infeasible with the current technique. I would recommend research on this topic in a preclinical setting before testing the system on actual patients (see further comments on page 3). Suggestion: ‘the performance of the CARLO device in terms of precision and accuracy needs to be (further) studied in a preclinical setting before it can be used in sites where direct visual feedback on its performance is impossible’

Page 3
- major comment: the mentioning of absence of control of the depth of the osteotomy between the protection of the retractors in the aperture and pterygoid region is very valuable. It should be further expressed in, amongst others, the abstract section that as this is a safety concern that should be addressed in the future.
- minor comment:  the judgement error of the surgeon during the pre-operative phase on where the osteotomy line should be seems odd. Especially the second comment in which the line was adjusted to avoid hitting the molar tips (during surgery, when you can’t properly see the tips, compared to the strong control the surgeon has during the planning phase on this part) seems not to be in line with clinical practice. To the readers this will be interpreted as uncareful planning since changing the osteotomy line of a preplanned case with surgical cutting guides during surgery is to be considered (very) rare.
- major comment: based on your abstract, it seems that the capabilities of the system are still overestimated. Any claims on precision and accuracy should be dramatically weakened, especially in the abstract. If you design an osteotomy line preoperatively, you have to adjust it during the surgery (regardless of its cause) and then claim precision based on matching the adjusted line to the postoperative results; it does not seem to be a very valuable addition. Especially not if any preclinical data supporting this statement or functioning as a benchmark is not presented elsewhere.

Page 4
- minor comment: the adjustments concerning the ‘straight line osteotomy’ are well accepted. I do believe that the overall image that is created, for example in the abstract, regarding a ‘Le Fort’ or ‘Midface’ osteotomy performed with a laser are somewhat overstating the actual role of the laser during the surgery.
- major comment: the lack of data regarding CARLO’s performance in preclinical studies needs attention. A clear statement is necessary regarding this missing data in the discussion of this paper, at least stating that data on this matter is present but has not been published. It is an odd order of sequence to first publish data on a (partially) performed osteotomy on an actual patient claiming its precision without properly studying (and preferably publishing) its capabilities beforehand (as you mention this has not been performed in a study setting). A study on the accuracy and precision of this system on a series of cadavers would be of scientific importance.

Page 5
- minor comment:  table 1: the disadvantage of the extra time necessary for the setup compared to the time saving of creating a, for example, cutting guide during the planning phase weighs extra because it costs the patient extra anesthesia time.
- the adjusted part on laser surgery characteristics is appreciated

Page 6
- minor comment: the conclusion on the cost-effectiveness part seems to lack a comparison to the gold standard in orthognathic surgery at this moment: virtual planning and 3d-printed wafers without using cutting guides. This should result in an extra disadvantage to the system where, compared to the above mentioned, extended pre-operative planning of the case is still necessary (i.e. removal of any scattering on the osteotomy line)
- minor comment: I would suggest to change the position of ‘(waferless)’ to the first time orthognathic surgery is mentioned in the paragraph

Author Response

Dear Sir or Madam,

Thank you for your time.

Kind regards

Matthias Ureel
